# *SF3B1^K700E* mutation in human embryonic stem cells causes aberrant expression of immune-related genes

Mahtab Dastpak[1,☯,¤a], Claudia A. Mimoso[2,☯,¤b], Moein Farshchian[3], Christina L. Paraggio[1,¤c], Claudia E. Leonard[1,¤d], Shanye Yin[1,¤e], Binkai Chi[1,¤f], Kelsey W. Nassar[4], R. Grant Rowe[5,6], Jiuchun Zhang[1], Zhonggang Hou[1,¤g], George Q. Daley[6], Karen Adelman[2,*], Robin Reed[1,†]

1 Department of Cell Biology, Harvard Medical School, Boston, Massachussetts, United States of America, 2 Department of Biological Chemistry and Molecular Pharmacology, Harvard Medical School, Boston, Massachussetts, United States of America, 3 Division of Oncology, Laboratory of Cellular Therapy, Department of Medical and Surgical Sciences for Children & Adults, University-Hospital of Modena and Reggio Emilia, Modena, Italy, 4 Division of Medical Oncology, Department of Medicine, University of Colorado Anschutz Medical Campus, Aurora, Colorado, United States of America, 5 Boston Children's Hospital, Harvard Medical School, Boston, Massachussetts, United States of America, 6 Dana-Farber Boston Children's Cancer and Blood Disorders Center, Boston, Massachussetts, United States of America

† Passed away in July 23, 2022.
☯ These authors contributed equally to this work.
¤a Current address: Cell Line and Cell Bank Development, Mammalian Platform, CMC, Sanofi, Framingham, Massachussetts, USA
¤b Current address: Yale School of Medicine, Department of Genetics, New Haven, Connecticut, USA
¤c Current address: Yale School of Medicine, Department of Microbial Pathogenesis, New Haven, Connecticut, USA
¤d Current address: Stanford University School of Medicine, Stanford, California, USA
¤e Current address: Department of Pathology, Albert Einstein College of Medicine, Bronx, New York, USA
¤f Current address: Leverna Therapeutics, Woburn, Massachussetts, USA
¤g Current address: Department of Biological Chemistry, University of Michigan, Ann Arbor, Michigan, USA
* karen_adelman@hms.harvard.edu

## Abstract

SF3B1, a component of the U2 snRNP pre-mRNA splicing factor, plays a critical role in splicing and is frequently mutated in cancer, particularly hematologic malignancies. We investigated the effects of the most common *SF3B1* mutation, heterozygous substitution of Lysine 700 to Glutamate (K700E), in human embryonic stem cells (hESC), using CRISPR-Cas9 to generate heterozygous *SF3B1^K700E* clones. Interestingly, we observed the upregulation of several key transcription regulators associated with hematopoiesis and a broad range of immune genes in *SF3B1^K700E* hESCs. Despite differences in the transcriptional and splicing profiles between hESC and myelodysplastic syndrome (MDS) cells harboring the *SF3B1^K700E* mutation, several common immune gene programs were identified in both cell types. To elucidate the molecular mechanisms underlying dysregulated gene expression in *SF3B1^K700E* hESCs, we mapped actively engaged RNA polymerase II (RNA Pol II) using Precision Run-On sequencing (PRO-seq). These analyses revealed that the *SF3B1^K700E* mutation alters RNA Pol II elongation properties. Specifically, we observed a general increase in

**Data availability statement:** RNA-seq and PRO-seq data reported in this paper were uploaded to Gene Expression Omnibus (GEO) database, accession number GSE278364.

**Funding:** This work was supported by NIH grant NIGMS GM122524 to RR, NIGMS GM139960 to KA, and Boston Children's Hospital Office of Faculty Development to RGR. No sponsor played any role in study design, data collection and analysis, decision to publish, or preparation of the manuscript.

**Competing interests:** I have read the journal's policy and the authors of this manuscript have the following competing interests: K.A. received research funding from Novartis not related to this work, consults for Syros Pharmaceuticals and Odyssey Therapeutics, and is on the SAB of CAMP4 Therapeutics. No aspect of this work is related to my consulting or other activities. This does not alter our adherence to PLOS ONE policies on sharing data and materials.

pause release in $SF3B1^{K700E}$ hESCs, consistent with recent work in leukemia cells suggesting that the $SF3B1^{K700E}$ mutation affects early transcription elongation. Taken together, our study identifies several candidate genes that could contribute to the $SF3B1$ mutated phenotype and clarifies the role for the U2 snRNP and pre-spliceosome assembly on transcription by RNA Pol II. Further, our data suggest that mutations of $SF3B1$ impact immune gene expression independent of cell type, providing new insights into the role of $SF3B1^{K700E}$ in hematologic malignancies.

## Introduction

SF3B1 is a component of the essential splicing factor U2 snRNP (small nuclear ribonucleoprotein) and is the most frequently mutated spliceosomal gene in cancer [1–3]. Pan-cancer genome analyses indicate that heterozygous $SF3B1$ mutations are primarily associated with hematological cancers, however, the fitness advantage these mutations provide to cancer cells is poorly understood. Pathogenic $SF3B1$ mutations are most frequent in myelodysplastic syndromes (MDS) and are also found in acute myeloid leukemia (AML), chronic lymphocytic leukemia (CLL) and several solid tumors, such as breast cancer (BRCA) and uveal melanoma (UVM) [4–9].

SF3B1 functions early in spliceosome assembly, and plays a central role in recognition of the 3' end of introns [10]. Previously, we cloned and characterized components of the U2 snRNP-associated complexes SF3a and SF3b [11–16]. Our studies, along with recent cryo-EM and crystallographic work, revealed that the SF3B1 component of SF3b binds to pre-mRNA on both sides of the branchpoint sequence (BPS) at the spliceosome core [15–20]. Mutations in $SF3B1$ cause a variety of mis-splicing events, leading to alterations in cellular pathways that may drive pathogenesis [7,21,22]. For example, recent studies revealed that mutant $SF3B1$ leads to mis-splicing of the NF-κB signaling inhibitor, $MAP3K7$, which was proposed to play a role in MDS pathogenesis [22–24]. However, another study reported that $MAP3K7$ mis-splicing leads to downregulation of $GATA1$, a master regulator of erythroid differentiation, and that this in turn results in the severe anemia associated with MDS [22–25].

To date, studies of cancer-associated $SF3B1$ mutations have been carried out in blood cancer patient cells (MDS, AML, CLL), blood cells from mouse models, blood cancer cell lines (e.g., NALM and K562), or non-blood transformed cell lines (e.g., HeLa and HEK293 cells) [23,26–33]. In our study, we sought to define the fundamental effects of the K700E mutant of $SF3B1$ by examining it in a non-transformed, ground state pluripotent cell. The $SF3B1^{K700E}$ mutation is the sole independent predictor of worse overall survival in MDS [34], with conditional knock-in mouse models developing macrocytic anemia, erythroid dysplasia, and long-term hematopoietic stem cell expansion [35]. Interestingly, this specific mutation also plays a tumorigenic role in other cancers, such as pancreatic ductal adenocarcinoma, suggestive of diverse roles in derivatives of multiple germ layers [36].

We CRISPR-edited hESCs to harbor a heterozygous K700E mutation and used RNA-seq to examine changes in gene expression and splicing. Surprisingly, several

master transcription regulators of hematopoiesis and numerous immune genes were upregulated in the *SF3B1^K700E^* hESCs, suggesting that wild-type SF3B1 plays a global role in ensuring proper regulation of immune genes. Intriguingly, in blood and non-blood cancers with the *SF3B1* mutation, immune pathways are among the top downregulated pathways. To gain a better understanding of the underlying molecular mechanisms, we employed nascent RNA analysis to examine the transcriptional landscape in *SF3B1^K700E^* hESCs. These analyses demonstrated that RNA Pol II elongation properties are altered in hESCs with the *SF3B1^K700E^* mutation, specifically showing a general increase in pause release.

## Materials and methods

### Cell culture

H9 hESC lines were cultured using the mTeSR1™1 kit (STEMCELL technology) and 1% penicillin-streptomycin (Sigma) on Matri-coated (CORNING) tissue culture plates and incubated at 37°C in humidified 5% (vol/vol) $CO_2$.

### CRISPR editing of *SF3B1^K700E^* in hESCs

H9 ES cells (WA09) obtained from WiCell Research Institute (Madison, WI) were cultured in E8 medium on Matrigel-coated plates at 37 °C with 5% (vol/vol) $CO_2$ [37]. CRISPR/CAS9 was used by the HMS Cell Biology Initiative for Genome Editing and Neurodegeneration to generate lines containing an A>G point mutation resulting in K700E amino acid substitution in the *SF3B1* gene. The CRISPR guide sequence was 5'-TGGATGAGCAGCAGAAAGTT-3'. The Ultramer sequence used to introduce the mutation was: 5'-TGTAACTTAGGTAATGTTGGGGCATAGTTAAAACCTGTGTTT GGTTTTGTAGGTCTTGTGGATGAGCAGCAGgAAGTTCGGACCATCAGTGCTTTGGCCATTGCTGCCTTGGCTGAA GCAGCAACTCCTTATGGTATCGAATCTTTTGAT-3 (point mutation lowercase; PAM region underscored). To create H9 cells harboring a heterozygous K700E mutation in *SF3B1*, 0.6 µg sgRNA was incubated with 3 µg SpCas9 protein for 10 minutes at room temperature and electroporated into $2x10^5$ H9 cells along with the Ultramer repair template. Mutants were identified by Illumina MiSeq. In addition, K700E mutation status was further confirmed by PCR amplification of genomic DNA, followed by Sanger sequencing using a primer set specific for *SF3B1*. The primer sequences were as follows: forward, 5'-AATTTGGGCTACTGATTTGGGGA-3'; reverse, 5'-GCCTTCAAGAAAGCAGCCAAAC-3'.

### Bulk RNA sequencing and data processing

Total RNA isolation from *SF3B1^K700E^* and WT hESCs was performed using the RNeasy Kit (Qiagen) following manufacturer's instructions. Bulk mRNA sequencing of triplicate samples was performed using NEBNext® Ultra™ II RNA Library preparation (NEB #E7775) with PolyA selection on the Illumina HiSeq4000 using the standard protocol for Illumina. On average, ~31M paired-end reads were obtained per sample.

All custom scripts described are available on the AdelmanLab GitHub (https://github.com/AdelmanLab/NIH_scripts). For Figs 2 and 6, the custom script (trim_and_filter_PE.pl) was used to trim FASTQ read pairs to 75 bp per mate and read pairs with a minimum average base quality score of 20 retained. Read pairs were further trimmed using cutadapt 1.14 to remove adapter sequences and low-quality 3' bases (--match-read-wildcards -m 20 -q 10). Reads were aligned to human (hg38) genome using parameters --quantMode TranscriptomeSAM GeneCounts --outMultimapperOrder Random --outSAMattrIHstart 0 --outFilterType BySJout --outFilterMismatchNmax 4 --alignSJoverhangMin 8 --outSAMstrandField intronMotif --outFilterIntronMotifs RemoveNoncanonicalUnannotated --alignIntronMin 20 --alignIntronMax 1000000 --alignMatesGapMax 1000000 --outFilterScoreMinOverLread 0 --outFilterMatchNminOverLread 0. Duplicates were also removed using STAR. Stranded coverage bedGraph files were generated from deduplicated BAM files using STAR. BedGraphs were depth-normalized using the normalize_bedGraph custom script. For Fig 2D, feature-counts was used to count RNA-seq reads within exons per gene and DESeq2 was run using default parameters to generate a list of differentially expressed genes between *SF3B1^K700E^* mutant and WT hESCs. For Fig 6G, depth-normalized

BedGraph files were converted to the bigWig format, and merged bedGraphs for each experimental condition were generated using bigWigMerge (UCSC tools). Merged bedGraphs were then converted to the bigWig format for visualization.

## Data processing of publicly available MDS and CLL patient data

Raw FASTQ files for MDS and CLL patients, K562 and NALM-6 cell lines, were downloaded from NCBI-GEO datasets (accession numbers GSE85712, GSE128805, GSE128429, GSE72790, GSE95011) (S1 Table). Data related to Fig 1 (hESC RNA-seq) and public datasets were aligned using STAR version 2.5.2b with the following parameters:: --outSAMtype BAM –SortedByCoordinate --outSAMunmapped Within --outSAMattributes Standard --quantMode GeneCounts [38]. Ensembl database (human release 99) and Human assembly hg38 (GRCh38) were used as the gene annotation and reference genome, respectively. In addition, TCGA-BRCA, TCGA-UVM, and AML from the BEATAML1.0-COHORT, including aligned BAM files and STAR 2-Pass Genome counts, were downloaded from the GDC data portal (https://portal.gdc.cancer.gov/).

## Identification of differential splicing events

To estimate alternative splicing (AS) events, PSI-Sigma v1.9j was used to provide a comprehensive analysis of AS events [39]. We used the default parameters of PSI-sigma, which required at least 10 supporting reads to detect splicing events with *Homo sapiens* GRCh38.99 as a reference genome. For GDC aligned BAM files, SJ.out files were generated. To quantify changes in splicing patterns, percent-spliced-in (PSI) index and differential PSI (ΔPSI) values were calculated to find all isoforms in a specific gene region. Splicing events with $\Delta PSI \geq 10\%$ and $p\text{-value} < 0.05$ were included in analyses. Additionally, Integrative Genomic Viewer (IGV_2.8.9) was used for visualizing aberrant splicing events in *SF3B1* mutant and wildtype samples.

## Gene set enrichment analysis (GSEA)

Gene ontology (GO) analysis was performed using GSEA (release v4.1.0) with gene set collection 'c5.go.bp.v7.4.symbols.gmt'. The pre-ranked gene list was analyzed to identify biological processes positively or negatively associated with *SF3B1* mutations. Gene sets with significant NES (NOM $p\text{-value} < 0.05$) were evaluated.

## Co-expression and weighted gene co-expression network analysis (WGCNA)

Weighted gene co-expression network analysis (WGCNA) was used to construct a transcriptional network from RNA expression profiles. WGCNA (R package, 'WGCNA' version 1.70–3) identifies the key modules and module-trait relationship using topological overlap measure (TOM) and detects similarly expressed gene sets across MDS samples [40].

## Precision run-on sequencing (PRO-seq)

The cell permeabilization was performed according to the protocols established by the Nascent Transcriptomics Core at Harvard Medical School, as described in Mimoso and Goldman [41]. Aliquots of frozen (−80°C) permeabilized cells were thawed on ice and pipetted gently to fully resuspend. Aliquots were removed and permeabilized cells were counted using a Luna FX7 (Logos Biosystems) instrument. For each sample, 1 million permeabilized cells were used for nuclear run-on with 50,000 permeabilized *Drosophila* S2 cells added for normalization. Nuclear run on assays and library preparation were performed as described in Mimoso and Goldman [41]. Pooled libraries were sequenced using the Illumina NovaSeq platform.

## PRO-seq data analysis

All custom scripts described herein are available on the AdelmanLab GitHub (https://github.com/AdelmanLab/NIH_scripts). Dual, 6 nt Unique Molecular Identifiers (UMIs) were extracted from read pairs using UMI-tools [42]. Read pairs were

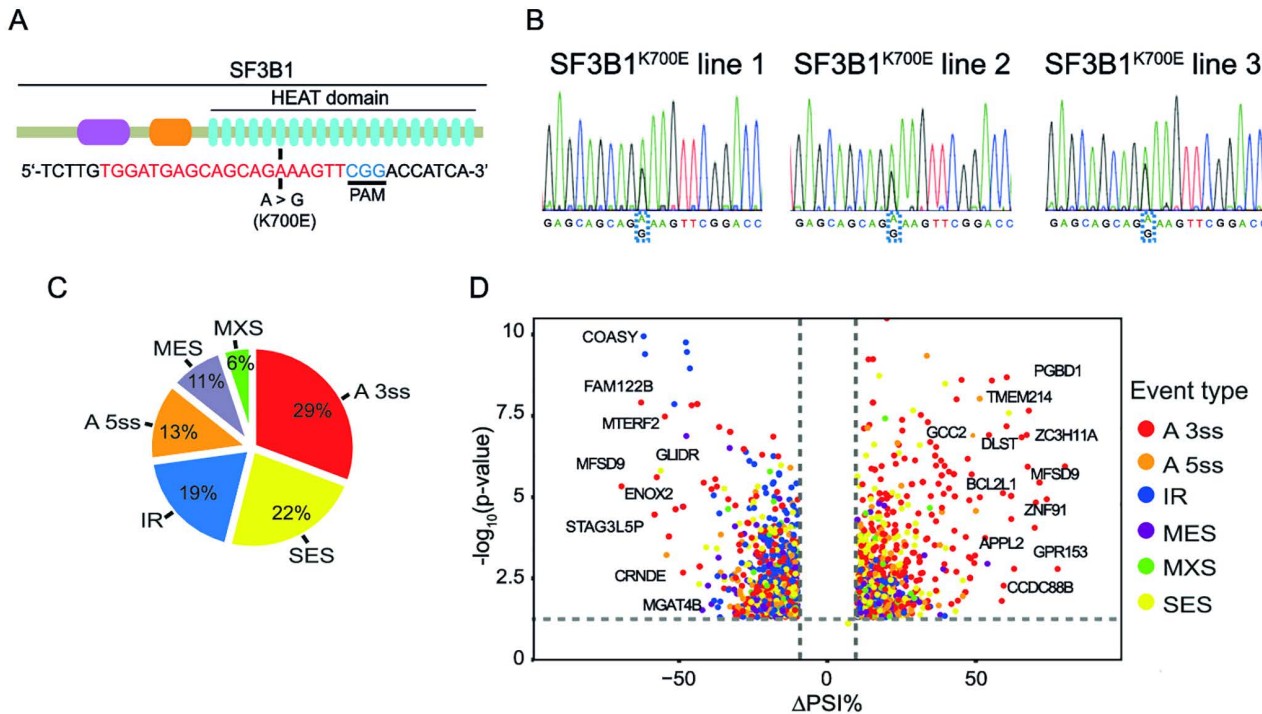

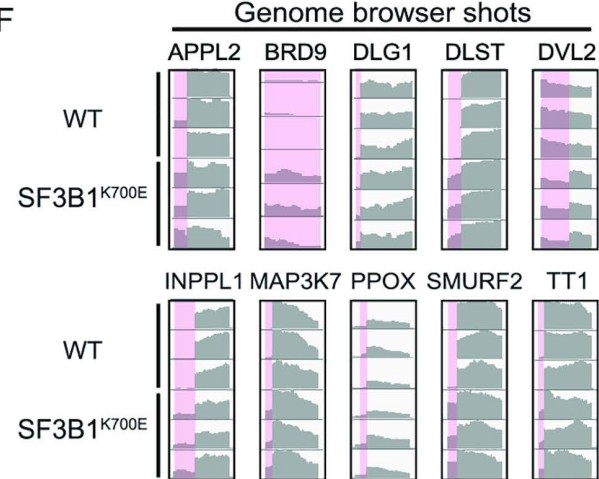

**Fig 1. CRISPR editing of the K700E mutation into one allele of *SF3B1* in human ES cells and the corresponding mis-splicing events. (A)** A schematic of *SF3B1* gene showing domain structure and position of K700E mutation. Target sequence for sgRNA is labeled in red, followed by PAM

sequence labeled in blue. **(B)** Sanger sequencing validation of heterozygous K700E mutation in three independent hESC clones. **(C)** Distribution of mis-splicing events in *SF3B1*<sup>K700E</sup> hESCs. ΔPSI ≥ 10% and p-value < 0.05 were used as thresholds. SES, single-exon skipping; MES, multiple-exon skipping; MXS, mutually exclusive splicing; A5ss, alternative 5′ splice site; A3ss, alternative 3′ splice site. **(D)** Volcano plot depicting mis-splicing events in *SF3B1*<sup>K700E</sup> ES cells. Representative genes with top ΔPSI% are labeled. **(E)** Representative mis-spliced genes in *SF3B1*<sup>K700E</sup> hESCs and *SF3B1*<sup>MT</sup> blood cancers (MDS, AML and CLL), blood cell lines (K562 and NALM6) and non-blood cancers (BRCA and UVM) with immune functions. **(F)** Integrated genome viewer (IGV) browser shots show mis-splicing in *SF3B1*<sup>K700E</sup> versus WT hESCs (highlighted in pink). Plots are from 3 biological replicates of an hESC line. See also S2 and S3 Tables.

trimmed using cutadapt 1.14 to remove adapter sequences (-O 1 --match-read-wildcards -m 26). The UMI length was trimmed off the end of both reads to prevent read-through into the mate's UMI, which will happen for shorter fragments. An additional nucleotide was removed from the end of read 1 (R1), using seqtk trimfq (https://github.com/lh3/seqtk), to preserve a single mate orientation during alignment. The paired end reads were then mapped to a combined genome index, including both the spike (dm6) and primary (hg38) genomes, using bowtie2 [43]. Properly paired reads were retained. These read pairs were then separated based on the genome (i.e., spike-in vs primary) to which they mapped, and both these spike and primary reads were independently deduplicated, again using UMI-tools. Reads mapping to the reference genome were separated according to whether they were R1 or R2, sorted via samtools 1.3.1 (-n), and subsequently converted to bedGraph format using a custom script (bowtie2stdBedGraph.pl). We note that this script counts each read once at the exact 3' end of the nascent RNA. Because R1 in PRO-seq reveals the position of the RNA 3' end, the "+" and "-" strands were swapped to generate bedGraphs representing 3' end positions at single nucleotide resolution.

Annotated transcription start sites were obtained from human (GRCh38.99) GTFs from Ensembl. After removing transcripts with {immunoglobulin, Mt, Mt_tRNA, rRNA} biotypes, PRO-seq signal in each sample was calculated in the window from the annotated TSS to +150 nt downstream, using a custom script, make_heatmap.pl. This script counts each read one time, at the exact 3' end location of the nascent RNA. Good agreement (spearman rho > 0.96) between replicates was observed.

BedGraphs were normalized using the normalize_bedGraph script. We observed a consistent increase in reads mapping to dm6 after introducing the *SF3B1*<sup>K700E</sup> mutation in hESCs (Average percentage of reads mapping to dm6: WT hESCs = 1.6%, *SF3B1*<sup>K700E</sup> hESCs = 2.1%), indicating a global decrease in transcription in *SF3B1*<sup>K700E</sup> mutant hESCs. As a result, the PRO-seq libraries were spike-in normalized. To generate the spike-in normalization factors per replicate, the number of reads mapping to dm6 for each sample was divided by the sample with the fewest number of reads mapping to dm6 (WT Replicate #1). bedgraphs2stdBedgraph was used to generate combined replicate bedGraphs by summing counts per nucleotide across replicates for each condition. BedGraphs were converted to the bigWig format for visualization.

An annotation of the single dominant transcription start site (TSS) and transcription end site (TES) per active gene in mESCs was obtained as described in Mimoso and Adelman 2023 with the following modifications: PRO-seq and RNA-seq data from WT and SF3B1<sup>K700E</sup> cells was used as input for the custom script GetGeneAnnotation (available on the Adelman Lab Github (get_gene_annotations.sh, https://github.com/AdelmanLab/GeneAnnotationScripts). The custom script make_heatmap was used to generate count matrices aligned to dominant TSSs [44]. Metagene plots were generated by summing reads within bins at each indicated position with respect to the TSS and dividing by the number of annotations. For all metagene plots, the bin size and number of investigated annotations are indicated in the figure legends. Box plots have a line at the median and whiskers depicting 10–90<sup>th</sup> percentile. GraphPad Prism was used to visualize metagene and box plots.

### Ethics statement

The patients' RNA-seq data presented in the current publication are based on the use of study data downloaded from the dbGaP website (S1 Table). This study does not meet the definition of human subjects research due to the fact that the data used was completely de-identified. Therefore, IRB approval was not required.

## Results and discussion

### The *SF3B1*<sup>K700E</sup> mutation results in specific RNA mis-splicing in hESCs

We established three independent heterozygous *SF3B1*<sup>K700E</sup> hESC lines using CRISPR/Cas9-mediated genome editing (Fig 1A), which were validated by Sanger sequencing (Fig 1B). To determine how the K700E mutation affected gene expression and splicing in hESCs we carried out RNA-seq on each of the three *SF3B1*<sup>K700E</sup> lines in triplicate, as compared to the wild type (WT) parental hESC line. These data confirmed that mutant and WT *SF3B1* alleles were expressed at similar levels in the *SF3B1*<sup>K700E</sup> hESC lines. Consistent with previous studies, the predominant mis-splicing event in *SF3B1*<sup>K700E</sup> hESCs was the use of cryptic, alternative 3' splice sites (Fig 1C and 1D; S2 Table). We observed mis-splicing of many genes, including *MAP3K7, DLST, DVL2, TMEM14C, DYNLL1, SNRPN,* and *BRD9*, that were reported in studies of mutant *SF3B1* cancers/ cell lines (Figs 1E and 1F) [9,23,28,35]. Moreover, the mis-splicing events occurred at identical coordinates in *SF3B1*<sup>K700E</sup> hESCs and SF3B1 cancers/transformed cell lines (Fig 1E; S3 Table).

### Immune genes are upregulated in *SF3B1*<sup>K700E</sup> hESCs

We next investigated the effect of the *SF3B1* mutation on gene expression in hESCs. Samples clustered well by genotype, with lines harboring the *SF3B1*<sup>K700E</sup> mutation clearly separating from the WT hESCs (Figs 2A-B). As shown in Fig 2C, we identified 1,177 up- and 577 down-regulated genes in *SF3B1*<sup>K700E</sup> hESCs (fold change ≥ 1.5, p-value < 0.05, S4 Table). Interestingly, several of the upregulated genes in the mutant hESC lines act as master regulators of hematopoiesis, with key roles in hematopoietic stem cells (HSCs), including *PITX2, ARID5B,* and *MAFB*. Additionally, we identified upregulated genes that contribute to the immune response [45–48]. These include *SERPINE* genes, which are serine protease inhibitors, and *GDF15*, a well-known marker for ineffective erythropoiesis (Fig 2D) [49,50].

To gain additional insight into the dysregulated genes in *SF3B1*<sup>K700E</sup> hESC lines, we carried out a gene set enrichment analysis (GSEA). This revealed that gene sets related to chromatin organization and DNA replication were downregulated, indicative of downregulation in *SF3B1*<sup>K700E</sup> hESC cells (S5 Table). These results are consistent with previous findings in cells treated with an inhibitor of SF3B1, which showed defective DNA replication and chromosome integrity [51]. Gene sets upregulated in *SF3B1*<sup>K700E</sup> hESCs were primarily associated with immune and inflammatory responses (Fig 2E, S5 Table), supporting a broad upregulation of related genes. Across all three lines, 18% of the total upregulated genes play roles in the immune response. A list of immune genes that were significantly upregulated in the *SF3B1*<sup>K700E</sup> hESC lines is shown in S6 Table. These data suggest that the essential splicing factor SF3B1 has a specific role in ensuring proper expression of hematopoietic and immune genes [22,35,52–54].

### Immune genes are downregulated in *SF3B1*<sup>MT</sup> MDS

Considering our observation that immune genes are upregulated in *SF3B1*<sup>K700E</sup> hESCs, we used GSEA to investigate gene expression in *SF3B1*<sup>MT</sup> MDS using published RNA-seq data (S1 Table) [21,35,55]. MDS patients who were WT for SF3B1 were used as controls. In striking contrast to hESCs bearing *SF3B1*<sup>K700E</sup> mutation, immune genes were typically downregulated in *SF3B1*<sup>MT</sup> MDS. In fact, a majority of the downregulated gene sets in *SF3B1*<sup>MT</sup> MDS samples are related to immunity (highlighted in green in Fig 3A; full list shown in S7 Table), suggesting broad downregulation of immune genes. Indeed, few immune gene sets were upregulated in *SF3B1*<sup>MT</sup> MDS (S7 Table). To further examine these unexpected findings, we applied weighted gene correlation network analysis (WGCNA) to identify gene networks correlated to *SF3B1*<sup>MT</sup> MDS gene expression [40]. Sixteen modules of densely interconnected genes were identified (Fig 3B; S2 Fig); the pathways within these modules are listed in S8 Table. High levels of similarity in the *SF3B1*<sup>MT</sup> MDS gene expression data sets are shown by the clustering of the 16 modules (Fig 3B) and by the eigengene adjacency heat map (Fig 3C). The M6_purple (r = 0.61, p-value = 3e-06) and M7_salmon (r = 0.31 and p-value = 0.03) modules, designated by the asterisks in Fig 3D, were negatively associated with immune pathways (Fig 3E,F). Among these are MHC class II pathways as well as

A

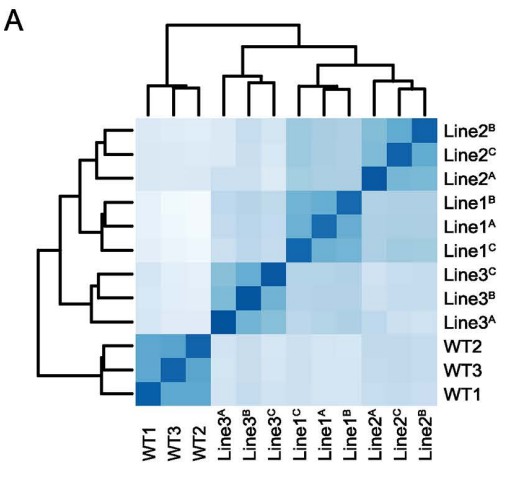

B

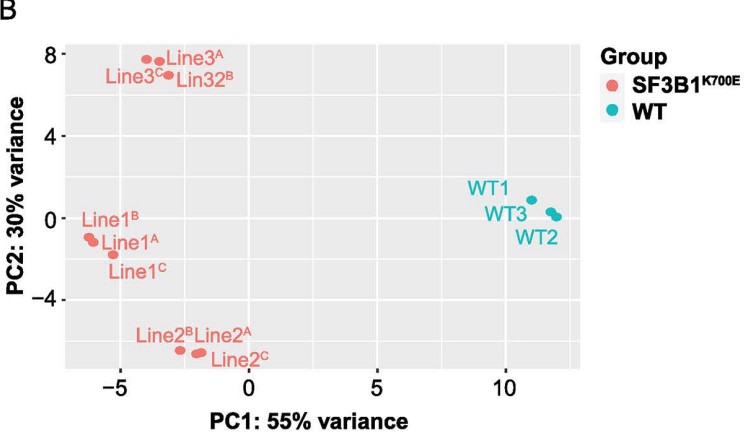

C

N=577   N=1,177

Padj (-log10) vs RNA-seq Fold Change SF3B1^{K700E/WT} (log2)

D

### Representative genes activated in SF3B1^{K700E} ES cells

| Gene | FC | Immune function |
|------|-----|-----------------|
| GDF15 | 24.5 | Cytokine associated with ineffective erythropoiesis |
| PITX2 | 14.2 | Self-renewal of HSCs/progenitors |
| ANXA1 | 6 | Key role in innate immune response |
| SERPINE1 | 6.1 | Affects immune cell infiltrations of cancers |
| SERPINB9 | 5.6 | Inhibits granzyme B |
| PLK2 | 4.1 | Role in innate immunity via TLR pathway |
| TAGLN | 3.6 | Functions in immunity & in cancer metastisis |
| TGM2 | 2.8 | Mediator of myeloid differentiation & role in leukemogenesis |
| MAFB | 2.7 | Regulates B cell development |
| ARID5B | 2.5 | Regulates lineage-specific hematopoiesis |

E

### Top 20 gene sets enriched in GSEA comparing WT to SF3B1^{K700E} ES cells

| NAME | NES |
|------|-----|
| GOBP_ANTERIOR_POSTERIOR_AXIS_SPECIFICATION | 2.01 |
| GOBP_GRANULOCYTE_ACTIVATION | 1.88 |
| GOBP_ACUTE_INFLAMMATORY_RESPONSE | 1.87 |
| GOBP_POSITIVE_REGULATION_OF_T_CELL_PROLIFERATION | 1.85 |
| GOBP_GENITALIA_DEVELOPMENT | 1.84 |
| GOBP_REGULATION_OF_COAGULATION | 1.83 |
| GOBP_REGULATION_OF_TYPE_2_IMMUNE_RESPONSE | 1.83 |
| GOBP_INTERLEUKIN_8_PRODUCTION | 1.82 |
| GOBP_STRIATED_MUSCLE_CELL_DEVELOPMENT | 1.81 |
| GOBP_POSITIVE_REGULATION_OF_LEUKOCYTE_CHEMOTAXIS | 1.81 |
| GOBP_REGULATION_OF_BIOLOGICAL_PROCESS_INVOLVED_IN_SYMBIOTIC_INTERACTION | 1.81 |
| GOBP_SINGLE_STRANDED_VIRAL_RNA_REPLICATION_VIA_DOUBLE_STRANDED_DNA_INTERMEDIATE | 1.80 |
| GOBP_POSITIVE_REGULATION_OF_INFLAMMATORY_RESPONSE | 1.80 |
| GOBP_DEFENSE_RESPONSE_TO_FUNGUS | 1.79 |
| GOBP_GRANULOCYTE_CHEMOTAXIS | 1.79 |
| GOBP_ACUTE_PHASE_RESPONSE | 1.79 |
| GOBP_POSITIVE_REGULATION_OF_COAGULATION | 1.79 |
| GOBP_DNA_DAMAGE_RESPONSE_SIGNAL_TRANSDUCTION_BY_P53_CLASS_MEDIATOR_RESULTING_IN_CELL_CYCLE_ARREST | 1.79 |
| GOBP_TYPE_2_IMMUNE_RESPONSE | 1.78 |
| GOBP_POSITIVE_REGULATION_OF_CD4_POSITIVE_ALPHA_BETA_T_CELL_ACTIVATION | 1.78 |

**Fig 2. Gene expression analysis in 3 independent *SF3B1^K700E* hESCs. (A)** Distance heatmap of RNA-seq data from *SF3B1^K700E* and WT ES cells in 3 biological replicates. **(B)** PCA plot of samples from WT and *SF3B1^K700E* mutant hES cells. **(C)** Volcano plot depicting differentially expressed genes (among N = 14,946 active genes) in *SF3B1^K700E* mutant vs. WT cells. Affected genes were defined by DESeq2 (p < 0.05 and Fold Change > 1.5). **(D)** Several representative master hematopoietic genes and immune genes FC: Fold Change. **(E)** Top 20 gene sets with the most positive normalized enrichment scores (NES), indicative of gene upregulation in *SF3B1^K700E* ES lines (NOM p-value < 0.05). See also S1 Fig and S4,S6 Tables.

pathways related to T and B cells (list of gene interactions see S9 Table). Together, these results support the conclusion that immune genes are downregulated in *SF3B1^MT* MDS. We considered that this could be a direct effect of *SF3B1* mutation, or an indirect effect of perturbed immune cell differentiation in *SF3B1*-mutated progenitors.

## Immune gene sets are broadly downregulated in *SF3B1^MT* cancers

We also carried out GSEA of *SF3B1^MT* CLL and AML using published datasets with the corresponding WT splicing factor patient data as controls (S1 Table) [28,56]. As observed with *SF3B1^MT* MDS, the strong negative enrichment of immune gene sets implies broad gene downregulation in AML or CLL-bearing mutant *SF3B1* (Fig 4A; S10 Table). We conclude that immune gene sets/genes are consistently downregulated in multiple different *SF3B1^MT* blood cancers.

As *SF3B1* mutation is also associated with non-blood cancers, we next asked whether the strong negative enrichment of immune gene sets in MDS/CLL/AML was due to cell type. Accordingly, we carried out GSEA using two *SF3B1* mutant non-blood cancers, breast cancer (BRCA) and uveal melanoma (UVM). Strikingly, we obtained the same results with these cancer cell types as with the blood cancers. The immune gene sets with significant enrichment scores in comparisons of *SF3B1^MT* versus *SF3B1^WT* BRCA and UVM exhibit negative associations, indicating downregulation of immune genes in *SF3B1^MT* solid cancers (Figs 4B-D, S11 Table).

Together, our data indicate that, although the mis-splicing patterns are similar between *SF3B1^K700E* ES cells and *SF3B1^MT* cancers, there is a striking difference in expression of the immune gene sets in *SF3B1^MT* ES cells versus cancer cells. Consistent with previous studies, we conclude that the function of the *SF3B1^K700E* mutation in ES or other cancers will be affected by cell state, as well as genomic and transcriptomic background [57].

## Downregulation of genes associated with leukocyte migration is common among SF3B1 cancers and U2AF1, SRSF2, and ZRSR2-mutated MDS

To identify common hallmarks between different damaging mutations in splicing factors in a variety of cancers and MDS, we examined whether any of the GSEA immune gene sets overlapped in the public datasets of blood and non-blood cancers associated with *SF3B1* mutations. This analysis revealed 7 gene sets in common. Among them were lymphocyte pathways and leukocyte migration (Fig 5A). We then examined GSEA data for *SRSF2^MT* and *ZRSR2^MT* MDS. Mutations in *SRSF2*, which binds to exonic splicing enhancers, alter exon inclusion [58]. ZRSR2 is mainly associated with the minor spliceosome, and its mutation primarily results in intron retention [59]. Remarkably, the leukocyte migration gene set was also downregulated in these splicing factor-mutated cancers (Fig 5B). Moreover, this gene set was the only one in common among all the splicing factor cancers that we examined, including *SF3B1^MT* MDS, AML, CLL, UVM and BRCA as well as MDS associated with mutations in SRSF2, ZRSR2 or U2AF1, a protein that interacts with U2AF2 to recognize the 3' splice site [58–60] (Fig 5B).

Additionally, at the level of individual genes, we identified one gene, *CCR1*, that is downregulated in all 5 *SF3B1^MT* cancers examined in this study. CCR1 is one of ~20 C-C motif chemokine receptors (CCRs), which are 7 membrane proteins that couple to G proteins for signal transduction [61]. CCRs are key regulators of migration and chemotaxis of multiple blood cell types, including thymocytes, granulocytes, myeloid cells, and additional types of leukocytes. We observed downregulation of *CCR1* by as much as 256-fold in CLL to ~2–3 fold in other *SF3B1^MT* cancers (S12 Table). Notably, *CCR1* knockout mice show impaired hematopoiesis as well as defective immune and inflammatory responses [62]. The

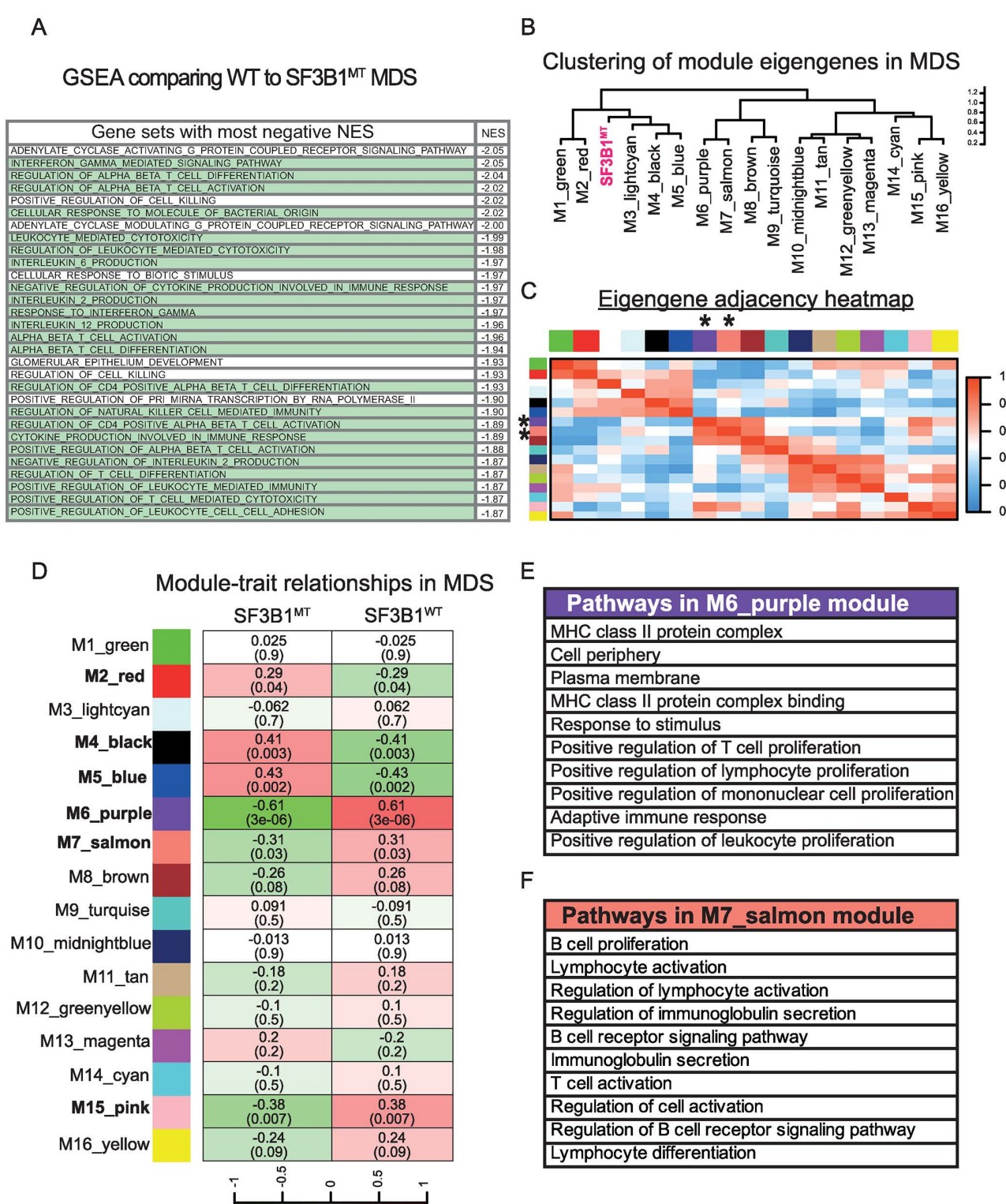

**Fig 3. Hematopoietic/Immune gene sets are generally downregulated in *SF3B1^MT* MDS. (A)** Gene set enrichment analysis, showing the most downregulated gene sets in *SF3B1^MT* MDS patients. Hematopoietic/Immune pathways are highlighted in green. Gene sets with significant NES (NOM p-value < 0.05) are shown. **(B)** Gene co-expression network in *SF3B1^MT* patients using WGCNA shows clustering of 16 module eigengenes in MDS

patients. *SF3B1^MT^* module is labeled in pink. **(C)** Eigengene adjacency heatmap indicates the correlation between modules. Red and blue represent positive and negative correlations, respectively. Asterisks indicate modules negatively associated with immune pathways. **(D)** Heatmap of module-trait relationship displaying correlation coefficients between module eigengenes and *SF3B1^MT^* phenotype. Colors indicate positive (red) or negative (green) correlations. Values without parentheses indicate correlation coefficient, corresponding p-values are in parentheses at the bottom of each row. Modules with significant p-values are in bold. **(E, F)** Enriched pathways in modules M6_purple and M7_salmon, respectively, are listed. See also S2 Fig and S7-S9 Tables.

potential importance of *CCR1* in splicing factor-mutant cancers is further suggested by the observation that this gene is downregulated in *U2AF1^MT^* MDS (Fig 5C). Moreover, in *SRSF2* and *ZRSR2* mutant MDS, *CCR2* is downregulated, and this CCR may substitute for the downregulation of *CCR1* as both receptors share a ligand (CCL7). In general, CCRs may play an important role in cancers associated with splicing factor mutations, since additional CCRs are downregulated in SF3B1 blood and non-blood cancers, as well as U2AF1 MDS, with downregulation ranging from 2 to 50-fold (Fig 5C; S12 Table).

The CCRs have multiple ligands which are usually C-C motif chemokine ligands (CCLs). We found that many of the CCLs are also downregulated in *SF3B1^MT^* cancers, and in other cancers associated with splicing factor mutations (Fig 5D; S12 Table). For example, the CCL ligands for CCR1 that are downregulated in *SF3B1* MDS include *CCL3*, *CCL4*, *CCL5*, *CCL7*, *CCL14* and *CCL23* (Fig 5D). Critically, although SF3B1 is best known for its role in splicing, we did not detect mis-splicing of the *CCR/CCLs* (S2 Table), suggesting that *SF3B1* mutations might also impact the process of transcription, 3' end formation or RNA decay.

## Splicing factor mutation increases transcriptional pause release in hESCs

Analyses of gene activity in *SF3B1^MT^* cells, whether in hESCs or cancer cells, have demonstrated that changes in gene expression are not merely the consequence of alterations in splicing (S2 and S4 Tables). This intriguing observation suggests a potential influence of the *SF3B1^K700E^* mutation on the process of transcription. To test this idea directly, we performed PRO-seq, which maps the position of engaged RNA Pol II at single nucleotide resolution [41,63] This involves the transcriptional incorporation of a biotin-NTP, which is used to halt transcription and stringently isolate nascent RNAs. To allow for absolute quantification of differences between WT and *SF3B1^K700E^* hESCs, exogenous RNAs were spiked into each sample.

We first investigated all genes differentially expressed in RNA-seq (as in Fig 2C), comparing the fold changes in RNA-seq vs. PRO-seq signal between *SF3B1^K700E^* and WT cells. This analysis revealed a strong positive correlation between the two datasets (Fig 6A, r = 0.66), indicating that the observed changes in steady state RNA levels largely reflect altered transcription. This result is consistent with the sizeable changes in gene expression observed in *SF3B1^K700E^* mutant cells, despite only modest splicing defects, and is in agreement with previous work showing that the U2 snRNP can influence transcription [64,65]. To probe this result further, we evaluated PRO-seq signal at several example genes that were differentially expressed genes in RNA-seq. At upregulated gene *FDXR*, we observed elevated signal in both RNA-seq and PRO-seq (Fig 6B, left), and at downregulated gene *MAP3K7*, we observed reduced signal in both assays (Fig 6B, right). Indeed, investigation of all upregulated genes showed a significant increase in gene body PRO-seq signal (Fig 6C), in agreement with the RNA-seq. Similarly, we observe a significant decrease in gene body PRO-seq signal at genes downregulated in RNA-seq (Fig 6D). This analysis confirms that the *SF3B1^K700E^* mutation elicits changes in transcription that contribute to altered steady state RNA levels.

We next wished to determine the mechanism by which genes are differentially transcribed in *SF3B1^K700E^* mutant cells. Interestingly, we observed a significant decrease in the peak of promoter RNA Pol II occupancy at both up- and down-regulated genes in *SF3B1^K700E^* mutant cells compared to WT (Fig 6E). Notably, this result is consistent with previous work evaluating changes in RNA Pol II elongation properties in *SF3B1^K700E^* knock-in K562 cells [66], which also reported

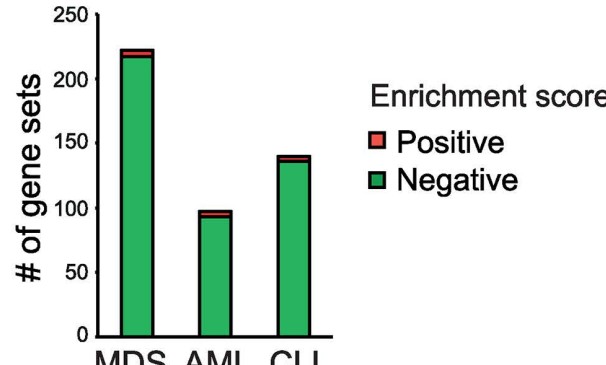

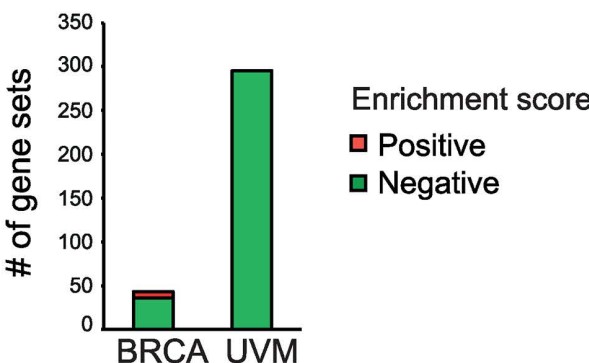

**A** GSEA comparing WT to SF3B1$^{MT}$ blood cancer

Immune-related gene sets

**B** GSEA comparing WT to SF3B1$^{MT}$ non-blood cancer

Immune-related gene sets

**C** GSEA comparing WT to SF3B1$^{MT}$ BRCA

| Gene sets with most negative NES | NES |
|---|---|
| IMMUNOGLOBULIN_PRODUCTION | -2.00 |
| HUMORAL_IMMUNE_RESPONSE_MEDIATED_BY_CIRCULATING_IMMUNOGLOBULIN | -1.95 |
| COMPLEMENT_ACTIVATION | -1.89 |
| REGULATION_OF_HUMORAL_IMMUNE_RESPONSE | -1.87 |
| HUMORAL_IMMUNE_RESPONSE | -1.80 |
| PRODUCTION_OF_MOLECULAR_MEDIATOR_OF_IMMUNE_RESPONSE | -1.77 |
| B_CELL_RECEPTOR_SIGNALING_PATHWAY | -1.76 |
| REGULATION_OF_COMPLEMENT_ACTIVATION | -1.74 |
| B_CELL_MEDIATED_IMMUNITY | -1.67 |
| ANTIBACTERIAL_HUMORAL_RESPONSE | -1.66 |
| POSITIVE_REGULATION_OF_B_CELL_ACTIVATION | -1.66 |
| ANTIMICROBIAL_HUMORAL_IMMUNE_RESPONSE_MEDIATED_BY_ANTIMICROBIAL_PEPTIDE | -1.65 |
| REGULATION_OF_B_CELL_ACTIVATION | -1.60 |
| ANTIMICROBIAL_HUMORAL_RESPONSE | -1.59 |
| INNATE_IMMUNE_RESPONSE_IN_MUCOSA | -1.54 |
| REGULATION_OF_B_CELL_PROLIFERATION | -1.53 |
| LYMPHOCYTE_MEDIATED_IMMUNITY | -1.53 |
| B_CELL_ACTIVATION | -1.51 |
| B_CELL_PROLIFERATION | -1.50 |
| CHRONIC_INFLAMMATORY_RESPONSE | -1.48 |
| ADAPTIVE_IMMUNE_RESPONSE_BASED_ON_SOMATIC_RECOMBINATION_OF_IMMUNE_... | -1.48 |
| POSITIVE_REGULATION_OF_MONOCYTE_CHEMOTAXIS | -1.45 |
| REGULATION_OF_LYMPHOCYTE_ACTIVATION | -1.27 |
| REGULATION_OF_IMMUNE_EFFECTOR_PROCESS | -1.24 |
| LEUKOCYTE_MIGRATION | -1.23 |
| PHAGOCYTOSIS_RECOGNITION | -1.81 |
| NEGATIVE_REGULATION_OF_EXECUTION_PHASE_OF_APOPTOSIS | -1.70 |
| DNA_METHYLATION_INVOLVED_IN_GAMETE_GENERATION | -1.68 |
| DNA_REPLICATION_DEPENDENT_NUCLEOSOME_ORGANIZATION | -1.68 |
| DOPAMINERGIC_NEURON_DIFFERENTIATION | -1.64 |

**D** GSEA comparing WT to SF3B1$^{MT}$ UVM

| Gene sets with most negative NES | NES |
|---|---|
| HUMORAL_IMMUNE_RESPONSE_MEDIATED_BY_CIRCULATING_IMMUNOGLOBULIN | -2.49 |
| COMPLEMENT_ACTIVATION | -2.45 |
| B_CELL_MEDIATED_IMMUNITY | -2.43 |
| PHAGOCYTOSIS_RECOGNITION | -2.42 |
| B_CELL_RECEPTOR_SIGNALING_PATHWAY | -2.40 |
| REGULATION_OF_HUMORAL_IMMUNE_RESPONSE | -2.40 |
| POSITIVE_REGULATION_OF_B_CELL_ACTIVATION | -2.40 |
| REGULATION_OF_COMPLEMENT_ACTIVATION | -2.38 |
| FC_RECEPTOR_MEDIATED_STIMULATORY_SIGNALING_PATHWAY | -2.31 |
| MEMBRANE_INVAGINATION | -2.31 |
| REGULATION_OF_B_CELL_ACTIVATION | -2.30 |
| IMMUNOGLOBULIN_PRODUCTION | -2.30 |
| FC_EPSILON_RECEPTOR_SIGNALING_PATHWAY | -2.28 |
| LYMPHOCYTE_MEDIATED_IMMUNITY | -2.27 |
| HUMORAL_IMMUNE_RESPONSE | -2.25 |
| ADAPTIVE_IMMUNE_RESPONSE_BASED_ON_SOMATIC_RECOMBINATION * | -2.24 |
| ANTIGEN_RECEPTOR_MEDIATED_SIGNALING_PATHWAY | -2.19 |
| FC_RECEPTOR_SIGNALING_PATHWAY | -2.16 |
| B_CELL_ACTIVATION | -2.14 |
| PHAGOCYTOSIS | -2.14 |
| PHOTOTRANSDUCTION | -2.13 |
| CELL_RECOGNITION | -2.12 |
| DEFENSE_RESPONSE_TO_BACTERIUM | -2.12 |
| IMMUNE_RESPONSE_REGULATING_SIGNALING_PATHWAY | -2.11 |
| PRODUCTION_OF_MOLECULAR_MEDIATOR_OF_IMMUNE_RESPONSE | -2.10 |
| REGULATION_OF_IMMUNE_EFFECTOR_PROCESS | -2.09 |
| POSITIVE_REGULATION_OF_CELL_ACTIVATION | -2.09 |
| LYMPHOCYTE_CHEMOTAXIS | -2.08 |
| RETINA_HOMEOSTASIS | -2.07 |
| DETECTION_OF_LIGHT_STIMULUS | -2.07 |

**Fig 4. Hematopoietic and immune gene sets are downregulated in *SF3B1*$^{MT}$ blood and solid cancers. (A, B)** Significant immune gene sets affected by *SF3B1* mutation in blood and non-blood cancers, respectively. The number of gene sets that are positively enriched (upregulated genes) as compared to downregulated genes are indicated. **(C, D)** Top gene sets with the most negative normalized enrichment scores (NES), indicative of gene downregulation in *SF3B1*$^{MT}$ BRCA and UVM patients, respectively. Hematopoietic/immune pathways are highlighted in green. Gene sets with significant NES (NOM p-value < 0.05) are shown. See also S7, S10, S11 Tables.

broadly lower RNA Pol II density near promoters. Thus, we wished to further probe the nature of this phenomenon. We considered that a reduction in promoter RNA Pol II levels might signify lower levels of transcription initiation in cells with the *SF3B1*$^{K700E}$ mutation, as suggested previously [66]. However, lower transcription initiation in *SF3B1*$^{K700E}$ cells would be expected to reduce gene expression and thus could not explain why most genes have unchanged or upregulated

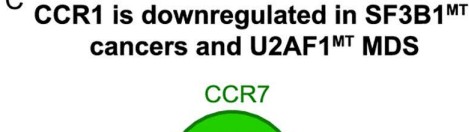

**A** **Shared immune gene sets negatively enriched in blood and non-blood cancers**

1) Regulation of lymphocyte activation
2) Regulation of immune effector process
3) Lymphocyte mediated immunity
4) Adaptive immune response based on somatic recombination of immune receptors built from immunoglobulin superfamily domains
5) Humoral immune response
6) Defense response to bacterium
7) Leukocyte migration

**B** **Leukocyte migration gene set is consistently downregulated in splicing factor-mutant cancers**

| | Datasets | NES | NOM p-value |
|---|---|---|---|
| SF3B1 | MDS | -1.4 | 0 |
| | AML | -1,7 | 0 |
| | CLL | -1.6 | 0 |
| | BRCA | -1.2 | 0.02 |
| | UVM | -2 | 0 |
| Other SFs | MDS-SRSF2 | -1.4 | 0 |
| | MDS-U2AF1 | -1.2 | 0 |
| | MDS-ZRSR2 | -1.9 | 0 |

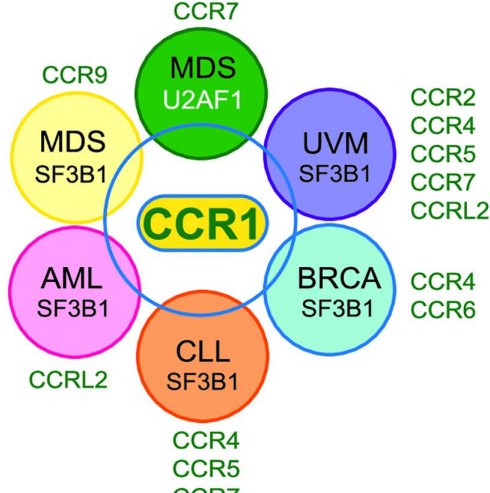

**C** **CCR1 is downregulated in SF3B1^MT cancers and U2AF1^MT MDS**

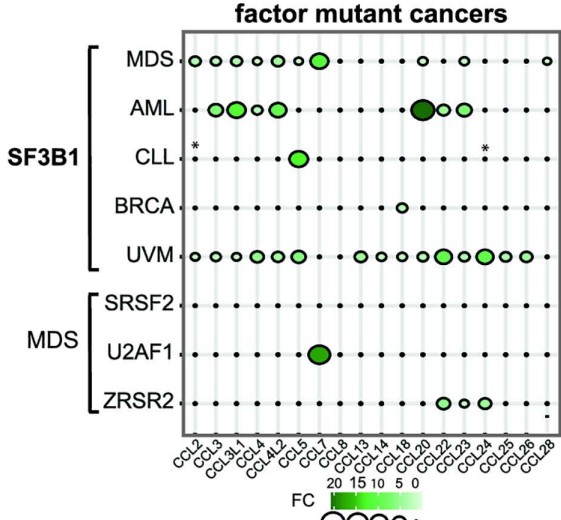

**D** **CCLs downregulated in splicing factor mutant cancers**

\* CCL2 (FC -254) and CCL24 (FC -6,884,402) in CLL are not in the plot due to their large FCs.

**Fig 5. Consistent downregulation of factors involved in signaling and migration in splicing factor mutant blood cancer. (A)** Shared immune GSEA gene sets downregulated in *SF3B1^MT* blood and non-blood cancers. Leukocyte migration gene set is highlighted in green. **(B)** Leukocyte migration is the only gene set shared among all splicing factors examined in this study. NES and NOM p-value are shown. **(C)** Venn diagram showing *CCR1* as the only shared downregulated gene among the indicated *SF3B1^MT* cancers and *U2AF1 ^MT* MDS. Additional CCRs present in the indicated cancers are shown. **(D)** Balloon plot showing downregulation of CCL genes in splicing factors (SF) cancers. See also S12 Table.

expression in *SF3B1^K700E* hESCs. We thus investigated whether the broadly lower levels of promoter-associated RNA Pol II suggests that the *SF3B1^K700E* mutation stimulates the release of paused RNA Pol II into productive elongation. To address this possibility, we calculated the pausing index, which is the ratio of promoter-proximal to gene body PRO-seq signal. Graphing the cumulative distribution of pausing indices in WT and *SF3B1^K700E* mutant cells revealed a significant reduction of pausing in cells with the *SF3B1^K700E* mutation (Fig 6F), at both upregulated (left) and downregulated genes (right). These results demonstrate that *SF3B1^K700E* mutant cells have more efficient pause release globally, which could explain the increased expression of upregulated genes. We envision that the hematopoietic and immune-related genes upregulated in *SF3B1^K700E* hESCs are normally rate limited at the level of pause release, such that a stimulation of pause release associated with the *SF3B1^K700E* mutation is sufficient to increase gene activity. In contrast, the downregulated

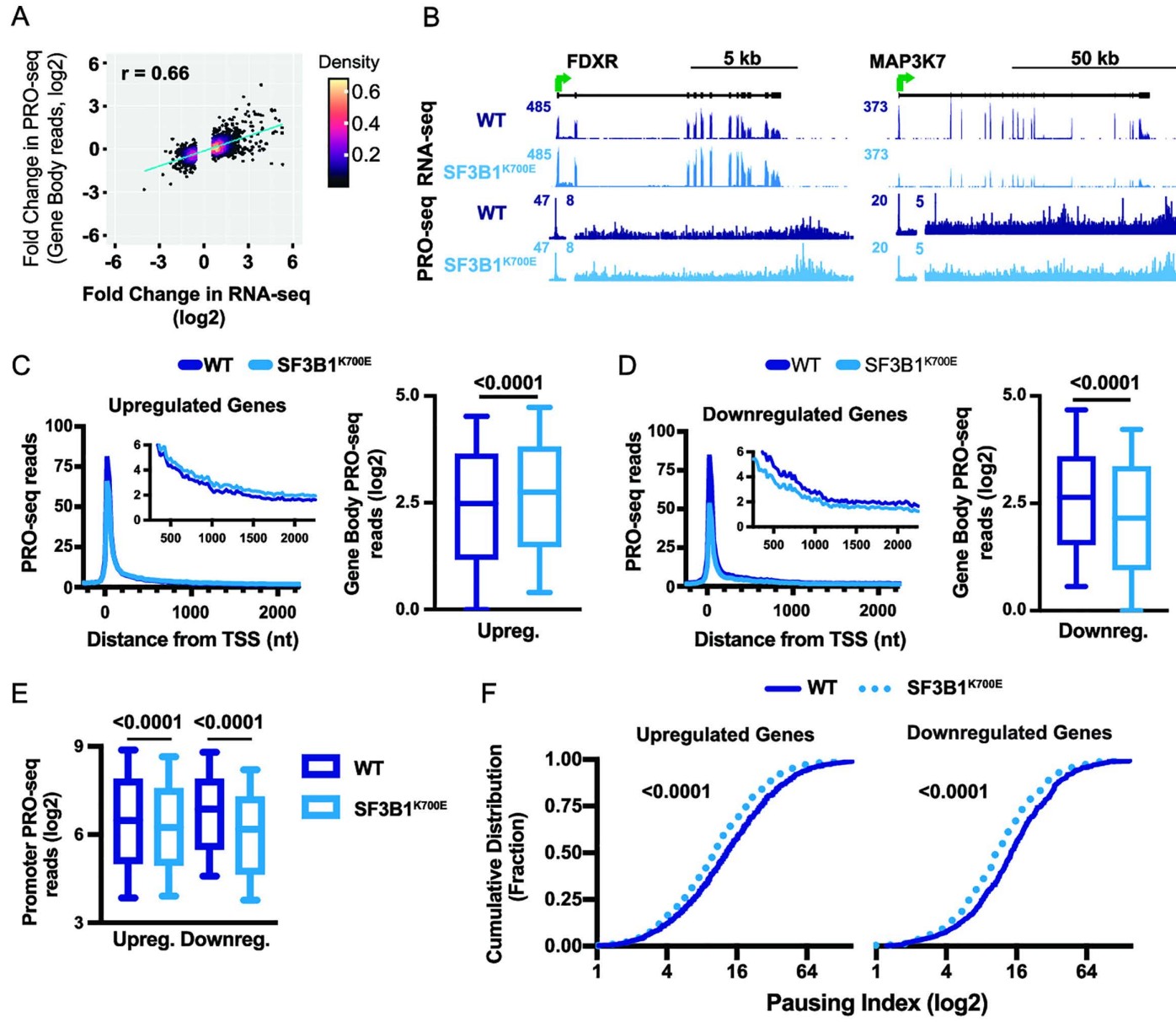

**Fig 6. *SF3B1^K700E* mutant cells exhibit altered transcription and more efficient pause release.** (A) Density scatter plot depicting the relationship between the fold change in RNA-seq and gene body PRO-seq signal (+250 to TES) at genes changed in RNA-seq (N = 1,754). **(B)** RNA-seq and PRO-seq at an example upregulated gene *FDXR* (left) and downregulated gene *MAP3K7* (right). PRO-seq signal was re-scaled downstream of the TSS to highlight gene body signal. **(C)** Metagene plot of PRO-seq signal at upregulated genes. Reads were summed in 25 nt bins, aligned to the TSS. Inset highlights gene body PRO-seq signal (TSS + 250 nt to + 2250 nt). At right are box plots reporting gene body PRO-seq reads (TSS + 250 to +2250nt) at upregulated genes per indicated condition. P-values generated using the paired t-test. **(D)** Same as C, but for downregulated genes. **(E)** Box plot reporting the fold change in promoter (TSS to +100 nt) PRO-seq signal at up- and down-regulated genes in WT and *SF3B1* mutant cells. P-values were calculated with the paired t-test. **(F)** Pausing index was calculated at upregulated and downregulated genes as the ratio of promoter to gene body PRO-seq read density. The cumulative distribution of pausing indices in *SF3B1^K700E* mutant and WT cells is shown. P-values generated using the paired t-test. See also **S13 Table**.

genes appear to have lower rates of transcription initiation, such that the RNA Pol II released from promoters is not effectively replaced by newly initiated RNA Pol II. In support of this idea, recent work in leukemia cells demonstrated that precocious pause release caused by *SF3B1* mutations could lead to reduced chromatin accessibility and epigenetic marks of transcription activity [66].

## Conclusions

The sophisticated interplay between transcription and splicing factors means that alterations in assembly of the spliceosome can have profound effects on the transcription process. Utilizing CRISPR-Cas9 technology, we developed a human embryonic stem cell (hESC) model with the *SF3B1^K700E^* mutation and assessed differential splicing, gene expression and nascent RNA synthesis. Our studies of SF3B1 function in a non-cancer cell line provides further insights into how this mutation alters the dynamics of RNA Pol II and mature mRNA output. We show that *SF3B1* mutations directly impact transcription, increasing pause release. We note that a recent study probing changes in RNA Pol II elongation after short-term inhibition of U2 snRNP with chemical inhibitors suggested that U2 snRNP stabilizes pausing, however our results and those from *SF3B1* mutant leukemia cell lines indicate that pause release is more efficient in cells with *SF3B1* mutations. Indeed, recent work has demonstrated that chemical inhibitors and splicing factor mutations elicit different effects on gene expression. Our analysis of patients' bulk RNA-seq data reflects a mixture of signals from different cell types, and future studies will be needed to assess gene expression changes within distinct cell populations from individuals with *SF3B1* mutations. Taken together with studies in cancer systems, our results support that formation of the pre-spliceosome including the U2 snRNP stimulates transcription elongation. We conclude that mutations in splicing factors have effects on RNA formation that go well beyond altered splicing profiles.

We suggest that more efficient pause release is a general feature of transcription in *SF3B1^K700E^* mutant cells, yet that the effect of the *SF3B1^K700E^* mutation on RNA levels can vary from cell type to cell type, dependent on whether transcription initiation or pause release is limiting at that gene. Certainly, the chromatin contexts of many genes differ between MDS compared to hESCs, which would undoubtedly influence the consequences of increased pause release. Nonetheless, we speculate that the similar mis-splicing patterns of *SF3B1^K700E^* mutation in hESCs and cancers cells reflect global roles of SF3B1 in the assembly and function of the pre-spliceosome.

Further investigation will be necessary to develop targeted strategies for inhibiting or ameliorating splicing-disrupted diseases such as MDS and AML. Among the different gene programs that could be exploited by cancer cells, mutation of splicing factors such as SF3B1 could increase the transcriptional output not only by affecting spicing itself but also by tuning RNA Pol II dynamics. Our results in hESC as an undifferentiated model showed that this specific mutation does not change the expression of pluripotency and stemness genes but drives a transcriptional signature related to hematopoiesis and immune responses. This finding supports the role of *SF3B1*^K700E^ in regulating immune signatures across multiple tumors and demonstrates that this could occur via a common mechanism impacting pausing by RNA Pol II. Future studies will demonstrate whether tumor-specific patterns of transcriptional dysregulation are controlled by this mechanism. While mutational landscapes in tumor or MDS samples may complicate the interpretation of the impact of *SF3B1* mutation on tumorigenesis, our hESC model could be useful to uncover the mechanisms underlying gene dysregulation in splicing factor-mutant cancers. Taken together, our study provides insights regarding the most common *SF3B1* mutation that might be applicable for future targeted therapeutics.

## Supporting information

**S1 Fig. Gene set enrichment analysis (GSEA) comparing _SF3B1_mutant versus WT ES cells.** (A) Bubble plot summarizing the top enriched biological pathways among differentially expressed genes. The x-axis shows the normalized enrichment score (NES), with positive values indicating upregulation in SF3B1_mutants and negative values indicating

downregulation. Bubble size reflects gene set size, and bubble color encodes the adjusted p-value (padj), only pathways with padj < 0.1 and |NES| > 1 are shown. (B) Representative enrichment plots for two leading pathways. Left, protein–DNA complex assembly (NES = −1.45; padj = 0.068) is negatively enriched in SF3B1 mutants. Right, regulation of inflammatory response (NES = 1.35; padj = 0.09) is positively enriched. For each plot, the running enrichment score (top, red) is displayed across the ranked gene list (middle barcode plot; red = up in SF3B1_mutant, blue = down), with the rank metric distribution shown below. (C) Bar plot of selected hallmark, GO, and Reactome pathways ranked by NES. Red bars show pathways upregulated in SF3B1_mutants; blue bars denote downregulated pathways. The x-axis shows NES.
(TIF)

**S2 Fig. Gene co-expression network analysis using WGCNA in *SF3B1^MU* MDS patients.** (A) Determination of soft-thresholding power in WGCNA. Scale-free topology index and mean connectivity for each power are shown. In this study, the threshold was reached for a power of 3. (B) Dendrogram of differentially expressed genes clustered based on the difference metrics (1-TOM**).** (C) Sample dendrogram and trait heatmap based on expression data from 3 MDS datasets.
(TIF)

**S1 Table. *SF3B1^MT* and *SF3B1^WT* patient information in GEO, TCGA and BeatAML cohorts.**
(XLSX)

**S2 Table. Mis-splicing events in RNA-seq from 3 biological replicates of *SF3B1^K700E* compared to *SF3B1^WT* ES cells.**
(XLSX)

**S3 Table. ΔPSI% and p-values for shared mis-splicing events in *SF3B1^K700E* ES cells and *SF3B1^MT* cancers/cell lines.**
(XLSX)

**S4 Table. Genes that are differentially expressed in *SF3B1^K700E* compared to *SF3B1^WT* ES cells.**
(XLSX)

**S5 Table. GSEA of dysregulated genes in *SF3B1^K700E* ES cells.**
(XLSX)

**S6 Table. Upregulated immune genes in *SF3B1^K700E* ES cells.**
(XLSX)

**S7 Table. GSEA analysis in *SF3B1^MT* and *SF3B1^WT* MDS patients.**
(XLSX)

**S8 Table. Enriched pathways in module eigengenes, related to Fig 3.**
(XLSX)

**S9 Table. Gene interactions in the M6_purple module and M7_salmon module.**
(XLSX)

**S10 Table. GSEA analysis in *SF3B1^MT* and *SF3B1^WT* AML and CLL patients.**
(XLSX)

**S11 Table. GSEA analyses of RNA-seq data in *SF3B1^MT* versus *SF3B1^WT* BRCA and UVM patients.**
(XLSX)

**S12 Table. Downregulated Chemokine Receptors (*CCRs*) and their ligands (CCLs) in splicing factor mutant cancers.**
(XLSX)

**S13 Table. RNA Pol II transcription levels measured using PRO-seq.**
(XLSX)

## Acknowledgments

We thank Dr. Thorsten M. Schlaeger for technical assistance with ES cell culture and characterization. We are grateful to Drs. John N. Hutchinson, Kuan-Ting Lin, and Research Computing Consultant groups at HMS for assistance in RNA-seq analysis. We thank WiCell Research Institute (Madison, WI) for H9 ES cells (WA09). The authors would like to thank the Nascent Transcriptomics Core at Harvard Medical School, Boston, MA for generating PRO-seq libraries. We thank Dr. Danesh Moazed for his critical comments on the manuscript.

## Author contributions

**Conceptualization:** Mahtab Dastpak, R. Grant Rowe, George Q Daley, Robin Reed.

**Data curation:** Mahtab Dastpak.

**Formal analysis:** Karen Adelman, Mahtab Dastpak, Claudia A Mimoso, Moein Farshchian.

**Funding acquisition:** Karen Adelman, Robin Reed.

**Investigation:** Karen Adelman, Mahtab Dastpak, Claudia A Mimoso, Christina L. Paraggio, Claudia E. Leonard, Shanye Yin, Binkai Chi, Kelsey W. Nassar, Jiuchun Zhang, Zhonggang Hou.

**Methodology:** Karen Adelman, Mahtab Dastpak, Claudia A Mimoso, R. Grant Rowe, George Q Daley.

**Project administration:** Karen Adelman, R. Grant Rowe, George Q Daley, Robin Reed.

**Resources:** R. Grant Rowe, George Q Daley, Robin Reed.

**Supervision:** Karen Adelman.

**Validation:** Mahtab Dastpak.

**Visualization:** Karen Adelman, Mahtab Dastpak, Claudia A Mimoso, Robin Reed.

**Writing – original draft:** Karen Adelman, Mahtab Dastpak, Claudia A Mimoso, Robin Reed.

**Writing – review & editing:** Karen Adelman, Mahtab Dastpak, Claudia A Mimoso, Moein Farshchian, Christina L. Paraggio, Claudia E. Leonard, Shanye Yin, Binkai Chi, Kelsey W. Nassar, R. Grant Rowe, Jiuchun Zhang, Zhonggang Hou, George Q Daley.

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
