## [Decision Letter · Decision Letter 0]

21 May 2025

Dear Dr. Adelman,

Thank you for submitting your manuscript to PLOS ONE. After careful consideration, we feel that it has merit but does not fully meet PLOS ONE’s publication criteria as it currently stands. Therefore, we invite you to submit a revised version of the manuscript that addresses the points raised during the review process, if possible.

We look forward to receiving your revised manuscript.

Kind regards,

Eric A. Shelden, Ph.D.

Academic Editor

PLOS ONE

Journal Requirements:

“This work was supported by NIH grant NIGMS GM122524 to RR, NIGMS GM139960 to KA, and Boston Children's Hospital Office of Faculty Development to RGR. We thank Dr. Danesh Moazed for his critical comments on the manuscript.”

“This work was supported by NIH grant NIGMS GM122524 to RR and NIGMS GM139960 to KA, and Boston Children's Hospital Office of Faculty Development to RGR.

no sponsor played any role in study design, data collection and analysis, decision to publish, or preparation of the manuscript?”

“I have read the journal's policy and the authors of this manuscript have the following competing interests: K.A. received research funding from Novartis not related to this work, consults for Syros Pharmaceuticals and Odyssey Therapeutics, and is on the SAB of CAMP4 Therapeutics.

no aspect of this work is related to my consulting or other activities”

Reviewers' comments:

Reviewer's Responses to Questions

**Comments to the Author**

1. Is the manuscript technically sound, and do the data support the conclusions?

Reviewer #1: Partly

Reviewer #2: Partly

Reviewer #3: Yes

2. Has the statistical analysis been performed appropriately and rigorously?

Reviewer #1: Yes

Reviewer #2: Yes

Reviewer #3: Yes

3. Have the authors made all data underlying the findings in their manuscript fully available?

Reviewer #1: Yes

Reviewer #2: Yes

Reviewer #3: Yes

4. Is the manuscript presented in an intelligible fashion and written in standard English?

Reviewer #1: Yes

Reviewer #2: Yes

Reviewer #3: Yes

Reviewer #1: The manuscript revealed that, in ES cell, SF3B1K700E mutation increases expressoion of transcription regulators associated with hematopoiesis and immune genes. Based on that the SF3B1 gene is a component of the U2 splicing factor, the authors elucidated the molecular mechanisms underlying the abberant expression of genes in SF3B1K700E mutated ES cell and the SF3B1 mutation altered RNA polymeraseII elongation, resulting in the abberant expression of immune-related genes. I wrote several comments below.

The authors claimed that we established three independent heterozygous clones; however, they did not provide how single-cell cloning was performed. Did the authors validate the heterozygosity of those ES clones?

Did the authors observe significant p-values and ΔPSI% for candidate genes in Figure 1E? Were the candidate genes in Figure 1E observed within the volcano plots in Figure 1D?

As FDXR was not mentioned earlier in the manuscript, why was it selected for investigating in Figure 6B instead of the previously examined genes? What is the significance of the FDXR gene? Is it related to the immune gene sets?

The authors concluded that the functional impact of SF3B1 K700E is modulated by cellular state and the underlying genomic–transcriptomic background, based on contrasting mis-splicing patterns in ES cell with gene-expression profiles from bulk MDS patient samples. However, these represent fundamentally different cellular mixtures, an undifferentiated single clone versus a heterogeneous population of mature and progenitor blood cells. To support the conclusion, it would be better that the authors would compare SF3B1 K700E effects on ES cell with either that on CD34-positive hematopoietic progenitor cells from patients, iPSC-derived CD34-positive cells or MDS/AML cell lines that have the SF3B1 mutation.

Reviewer #2: The manuscript under review described a new hESC model of SF3B1-K700E mutation. This is an important model as this would be characterizing the functional impact of SF3B1 mutation in a non-cancer cell model context. The model will be useful in the splicing field to assess the impact of SF3B1-K700E in normal cellular context. In this note, the authors found mis-splicing patterns that were similar to reported publications. One interesting aspect is the increased in inflammation signature in hESC K700E cells, which is not seen in cancer/disease models reports. The alteration in RNA Pol 2 elongation has been reported. This would be just validation in a different model system, which will still important and informative.

Couple suggestions that would strengthen the paper overall.

1. General comment regarding GSEA result. The tables in main figures to show signficant pathways up or down are not very informative without the actual NES and statistical values here (eventhough there's a full supplemental table). It would be more informative to create some kind of heatmaps to represent the overall GSEA data as well. It is a bit difficult to know how strong the correlations are without going through the supplemental tables.

2. Major criticism: There seems to be a disconnect between the increased inflammation signature, alternative splicing, and transcription. Can the authors evaluate whether the increased in inflammatory signatures is due to changes in alternative splicing of those genes? Are there any correlation there?

- Second, are inflammatory gens that are differentially expressed in Sf3B1-K700E hESC corrected with where the authors saw increased in pause release by PRO-seq? Are there any correlation here?

3. CCR1 - the observation that CCR1 is downregulated in different cancer types is interestingly, Can the authors comment on why this is? (alternative splicing, gene expression).

Missing references

- Seiler et al - PMID: 29617667 - also identified reduced inflamation signature in pan-cancer analysis. This should be consistent with the findings here and should be included.

Reviewer #3: The authors consider the effects of the SF3B1 splicing mutation on the expression, splicing and transcription of embryonic stem cells. They use CRISPR/Cas9 to base edit three different lines. They show that there is some agreement between the gene expression differences they see, And those observed in patient samples. The main interesting, if slightly confusing, finding of the paper is that while in the stem cells they see up regulation of immune genes, in patient samples, they see it up regulation of the same gene sets. Additionally, they generate data to measure PolII occupancy and conclude that the mutant cells have reduced PolII pausing.

Overall, I think this is a strong well written paper that easily meets the bar for a PLOS one publication. There is substantial experimental and computational work that went into the findings. My main concerns are around some of the conclusions that are made when analyzing public bulk RNA sequencing data sets. The fundamental problem is that they mix together the effects of cell type and self state proportion changes with cell intrinsic expression changes. This is less of an issue for the stem cell data the authors generated because it's just the one type. Indeed, it is possible that the down regulation of immune genes seen in the patient samples could be a result of changes in cell type proportion as a result of the mutation rather than changes within a single cell type. There is in fact, single cell long read RNA sequencing data from patients with SF3B1 mutations which could be used to potentially address this question: Mariela Cortés-López et al. 2023 "Single-cell multi-omics defines the cell-type-specific impact of splicing aberrations in human hematopoietic clonal outgrowths". Similarly for the breast cancer data one has to consider whether it is changes in the immune response that result in the observed differences between S3B1 mutant and wild type cancers.

If this were a higher tier journal, I would certainly be asking for the authors to look at the data from the paper above and also to try to find single cell data for some of the other cancers considered. For PLOS one I don't know if this should be required: personally I would be OK with an acknowledgment of these limitations of the analysis.

Minor comments:

"negatively enriched" is an awkward term which sounds very similar to "depleted", which is not what the authors mean. I would suggest using "downregulated".

p12l300: "networks" there are no networks here (yet)

p14l365: how much of the DGE in BRCA/UVA is immune related (and compared to blood cancer?)

**Do you want your identity to be public for this peer review?** For information about this choice, including consent withdrawal, please see our Privacy Policy

Reviewer #1: No

Reviewer #2: No

Reviewer #3: No

---

## [Author Response · Author response to Decision Letter 1]

14 Sep 2025

We thank the reviewers and editorial team for their constructive critiques. We have responded to each point raised, as described below. The changes made have been highlighted in the marked-up version of the manuscript, labeled as 'Revised Manuscript with Track Changes', and an unmarked, updated version has been uploaded as ‘Manuscript’.

The manuscript has been updated accordingly.

“This work was supported by NIH grant NIGMS GM122524 to RR, NIGMS GM139960 to KA, and Boston Children's Hospital Office of Faculty Development to RGR. We thank Dr. Danesh Moazed for his critical comments on the manuscript.”

Please remove any funding-related text from the manuscript and let us know how you would like to update your Funding Statement. Currently, your Funding Statement reads as follows: “This work was supported by NIH grant NIGMS GM122524 to RR and NIGMS GM139960 to KA, and Boston Children's Hospital Office of Faculty Development to RGR. No sponsor played any role in study design, data collection and analysis, decision to publish, or preparation of the manuscript?”

We have removed the funding information from the Acknowledgements.

As indicated in our cover letter, the funding statement should read: ““This work was supported by NIH grant NIGMS GM122524 to RR, NIGMS GM139960 to KA, and Boston Children's Hospital Office of Faculty Development to RGR. No sponsor played any role in study design, data collection and analysis, decision to publish, or preparation of the manuscript”

“I have read the journal's policy and the authors of this manuscript have the following competing interests: K.A. received research funding from Novartis not related to this work, consults for Syros Pharmaceuticals and Odyssey Therapeutics, and is on the SAB of CAMP4 Therapeutics. No aspect of this work is related to my consulting or other activities”

As noted in the cover letter, I confirm that I will adhere to all PLOS ONE policies on sharing data and materials. Thank you for changing the online submission to include the following statement: "This does not alter our adherence to PLOS ONE policies on sharing data and materials.”

The Ethics Statement has been added at the end of the Methods section in the manuscript file.

Given the unexpected passing away of Dr. Reed and the untimely closure of her lab, some of the original gel images were not available. We understand that we’d need to share the raw data if we include the western blots in Figure 1G in the previous version of the manuscript. Since we don’t have these images, we have removed panel 1G from the manuscript.

References have been updated.

Reviewers' comments:

Reviewer #1: The manuscript revealed that, in ES cell, SF3B1K700E mutation increases expressoion of transcription regulators associated with hematopoiesis and immune genes. Based on that the SF3B1 gene is a component of the U2 splicing factor, the authors elucidated the molecular mechanisms underlying the abberant expression of genes in SF3B1K700E mutated ES cell and the SF3B1 mutation altered RNA polymeraseII elongation, resulting in the abberant expression of immune-related genes. I wrote several comments below.

The authors claimed that we established three independent heterozygous clones; however, they did not provide how single-cell cloning was performed. Did the authors validate the heterozygosity of those ES clones?

For single cell cloning, transfected cells were sorted as single cells into 96-well plates. The 3 heterozygous clones generated were identified by MiSeq (see below).

We also did the Sanger sequencing to further confirm the heterozygosity, which was shown in Figure 1B.

Did the authors observe significant p-values and ΔPSI% for candidate genes in Figure 1E?

Yes. All data are presented in S4 Table: Significant ∆PSI% and p-values for shared mis-splicing events in SF3B1K700E ES cells and SF3B1MT cancers/cell lines.

Were the candidate genes in Figure 1E observed within the volcano plots in Figure 1D? All significant?

Yes. Figure 1D is using all ∆PSI ≥ 10% and p-value < 0.05. Only genes with top ΔPSI% were labeled.

As FDXR was not mentioned earlier in the manuscript, why was it selected for investigating in Figure 6B instead of the previously examined genes? What is the significance of the FDXR gene? Is it related to the immune gene sets?

FDXR was selected as an example gene to show the trend we report globally at the upregulated genes. This gene was not selected based on cherry picking a specific gene – it was just a randomly selected example to show the breadth of the phenomenon.

The authors concluded that the functional impact of SF3B1 K700E is modulated by cellular state and the underlying genomic–transcriptomic background, based on contrasting mis-splicing patterns in ES cell with gene-expression profiles from bulk MDS patient samples. However, these represent fundamentally different cellular mixtures, an undifferentiated single clone versus a heterogeneous population of mature and progenitor blood cells.

To support the conclusion, it would be better that the authors would compare SF3B1 K700E effects on ES cell with either that on CD34-positive hematopoietic progenitor cells from patients, iPSC-derived CD34-positive cells or MDS/AML cell lines that have the SF3B1 mutation.

To address the reviewer’s suggestion, we reanalyzed publicly available RNA-seq data from a study published in Blood Advances (2022) [DOI: 10.1182/bloodadvances.2021006325], which investigated the impact of the SF3B1K700E mutation in a clinically relevant context. In this study, the authors generated a panel of 16 genetically matched SF3B1K700E and SF3B1WT iPSC lines derived from patients with myelodysplastic syndromes with ring sideroblasts (MDS-RS) harboring isolated SF3B1K700E mutations. RNA-seq and ATAC-seq were performed on purified CD34⁺/CD45⁺ hematopoietic stem and progenitor cells (HSPCs) differentiated from these iPSC lines.

Our analysis results are summarized in S15 Table, which includes patient information, differential gene expression (DGE) data, shared mis-splicing events (related to S4 Table), gene set enrichment analysis (GSEA) outcomes, and chemokine receptor expression profiles (related to S13 Table). These findings are consistent with observations in SF3B1-mutant cancers, demonstrating recurrent mis-splicing events, downregulation of multiple chemokine receptors, and broad suppression of immune-related pathways as identified through GSEA.

Reviewer #2: The manuscript under review described a new hESC model of SF3B1-K700E mutation. This is an important model as this would be characterizing the functional impact of SF3B1 mutation in a non-cancer cell model context. The model will be useful in the splicing field to assess the impact of SF3B1-K700E in normal cellular context. In this note, the authors found mis-splicing patterns that were similar to reported publications. One interesting aspect is the increased in inflammation signature in hESC K700E cells, which is not seen in cancer/disease models reports. The alteration in RNA Pol 2 elongation has been reported. This would be just validation in a different model system, which will still important and informative.

Couple suggestions that would strengthen the paper overall.

1. General comment regarding GSEA result. The tables in main figures to show signficant pathways up or down are not very informative without the actual NES and statistical values here (even though there's a full supplemental table). It would be more informative to create some kind of heatmaps to represent the overall GSEA data as well. It is a bit difficult to know how strong the correlations are without going through the supplemental tables.

Figures 2, 3, and 4 have been updated to include the NES values.

2. Major criticism: There seems to be a disconnect between the increased inflammation signature, alternative splicing, and transcription. Can the authors evaluate whether the increased in inflammatory signatures is due to changes in alternative splicing of those genes? Are there any correlation there?

Interestingly, as has been reported by others, many of the gene expression changes in SF3B1 mutant cancer are not reflective of mis-splicing. We note this in the text (see below). This is precisely why we and others are beginning to evaluate a role for SF3B1 in transcription and RNA processing or decay.

413. CCL14 and CCL23 (Fig 5D). Critically, although SF3B1 is best known for its role in splicing, we

414. did not detect mis-splicing of the CCR/CCLs (S3 Table), suggesting that SF3B1

415. mutations might also impact the process of transcription, 3’ end formation or RNA decay.

- Second, are inflammatory gens that are differentially expressed in Sf3B1-K700E hESC corrected with where the authors saw increased in pause release by PRO-seq? Are there any correlation here?

No. No significant differential expression was observed for any of the CCR/CCL genes listed in S13 Table in SF3B1K700E hESC lines.

3. CCR1 – the observation that CCR1 is downregulated in different cancer types is interestingly, Can the authors comment on why this is? (alternative splicing, gene expression).

We appreciate the reviewer’s insightful comment. In our analysis, we did not detect any evidence of mis-splicing or alternative splicing events affecting CCR1 in either blood or non-blood cancer datasets included in this study. As reported in the main text and S13 Table, CCR1 was significantly downregulated across all cancer types analyzed. However, the specific mechanisms underlying this downregulation, whether due to transcriptional repression, epigenetic modifications, or post-transcriptional regulation, were not investigated in the current study and remain to be explored in future work.

Missing references

- Seiler et al – PMID: 29617667 – also identified reduced inflammation signature in pan-cancer analysis. This should be consistent with the findings here and should be included.

This reference has been added to the manuscript and listed as reference number 58 in the References section.

Reviewer #3: The authors consider the effects of the SF3B1 splicing mutation on the expression, splicing and transcription of embryonic stem cells. They use CRISPR/Cas9 to base edit three different lines. They show that there is some agreement between the gene expression differences they see, and those observed in patient samples. The main interesting, if slightly confusing, finding of the paper is that while in the stem cells they see up regulation of immune genes, in patient samples, they see it up regulation of the same gene sets. Additionally, they generate data to measure PolII occupancy and conclude that the mutant cells have reduced PolII pausing.

Overall, I think this is a strong well written paper that easily meets the bar for a PLOS one publication. There is substantial experimental and computational work that went into the findings. My main concerns are around some of the conclusions that are made when analyzing public bulk RNA sequencing data sets. The fundamental problem is that they mix together the effects of cell type and self state proportion changes with cell intrinsic expression changes. This is less of an issue for the stem cell data the authors generated because it's just the one type. Indeed, it is possible that the down regulation of immune genes seen in the patient samples could be a result of changes in cell type proportion as a result of the mutation rather than changes within a single cell type. There is in fact, single cell long read RNA sequencing data from patients with SF3B1 mutations which could be used to potentially address this question: Mariela Cortés-López et al. 2023 "Single-cell multi-omics defines the cell-type-specific impact of splicing aberrations in human hematopoietic clonal outgrowths". Similarly for the breast cancer data one has to consider whether it is changes in the immune response that result in the observed differences between S3B1 mutant and wild type cancers.

If this were a higher tier journal, I would certainly be asking for the authors to look at the data from the paper above and also to try to find single cell data for some of the other cancers considered. For PLOS one I don't know if this should be required: personally I would be OK with an acknowledgment of these lim

---

## [Editor Report · Decision Letter 1]

25 Sep 2025

SF3B1K700E mutation in human embryonic stem cells causes aberrant expression of immune-related genes

PONE-D-25-20170R1

Dear Dr. Adelman,

We’re pleased to inform you that your manuscript has been judged scientifically suitable for publication and will be formally accepted for publication once it meets all outstanding technical requirements.

Kind regards,

Eric A. Shelden, Ph.D.

Academic Editor

PLOS ONE
---

## [Editor Report · Acceptance letter]

PONE-D-25-20170R1

PLOS One

Dear Dr. Adelman,

I'm pleased to inform you that your manuscript has been deemed suitable for publication in PLOS One. Congratulations! Your manuscript is now being handed over to our production team.

Kind regards,

on behalf of

Dr. Eric A. Shelden

Academic Editor

PLOS One